# Radiation damping strongly perturbs remote resonances in presence of homo-nuclear mixing

Philippe Pelupessy[1]

[1]Laboratoire des Biomolécules, LBM, Département de Chimie, École Normale Supérieure, PSL University, Sorbonne Université, CNRS, 75005 Paris, France

**Correspondence:** Philippe Pelupessy (philippe.pelupessy@ens.psl.eu)

**Abstract.** In this work, it is experimentally shown that the weak oscillating magnetic field (known as the "radiation damping" field) caused by the inductive coupling between the transverse magnetization of nuclei and the radio frequency circuit perturbs remote resonances when homo-nuclear total correlation mixing is applied. Numerical simulations are used to rationalize this effect.

## 1   Introduction

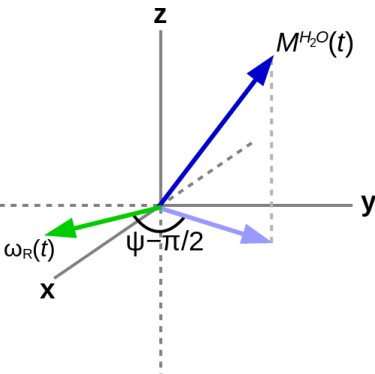

**Figure 1.** The RD field $\omega_R(t)$ (green arrow) lies in the $xy$-plane. It has an amplitude that is proportional to the projection (light blue arrow) of the water magnetization $M^{H_2O}(t)$ (dark blue arrow) onto the same plane and a phase of $\psi - \pi/2$ with respect to this projection.

The inductive coupling between precessing magnetization and a radio-frequency (RF) circuit creates an RF field, which in turn affects the evolution of the magnetization and hence the appearance of NMR spectra. The existence of this phenomenon was first hypothesized by Suryan (Suryan (1949)), while a more rigorous theoretical description was later provided by Bloembergen and Pound (Bloembergen and Pound (1954)). The latter introduced the term radiation damping (RD), an expression which, as several authors have stated before (Abragam (1961); Vlassenbroek et al. (1995); Hoult and Bhakar (1997); Krishnan

and Murali (2013)), is rather misleading with both radiation and damping called into question. The expression "Radiation feedback" has been suggested as an alternative, however this term is often used to designate active feedback circuits to enhance (Szoke and Meiboom (1959); Hobson and Kaiser (1975)) or eliminate (Louisjoseph et al. (1995); Broekaert and Jeener (1995)) the effects of radiation damping. Another option, in analogy with quantum backaction, could be *induction backaction*. In order to avoid confusion, the term RD will nevertheless be employed in this work. When no other RF fields are present, the RD field rotates the magnetization that is responsible for the induced RF field towards its equilibrium direction (Bloom (1957)), parallel to the main field, leaving the norm unchanged (if it is homogeneous in space). In liquid state NMR, this effect is usually weak and only noticeable when the magnetization is strong, either for nuclei in molar concentrations with high gyromagnetic ratios (in particular, solvents containing hydrogen) or when the polarization is enhanced. It increases with higher quality factors $Q$.

The RD field strongly affects the resonances with frequencies close to the one that is at its origin (Schlagnitweit et al. (2012)) and remote resonances that are directly coupled by scalar or dipolar interactions (Miao et al. (1999)) or undergo chemical exchange (Chen and Mao (1997)) with the nuclei that induce RD. Subtle effects on remote resonances (Sobol et al. (1998)) can affect sensitive difference experiments. Homo-nuclear isotropic mixing sequences which have been designed for total correlation spectroscopy (TOCSY), however, are very efficient at removing the chemical shift differences from the effective Hamiltonian (Braunschweiler and Ernst (1983); Bax and Davis (1985)). In this work, it will be shown that RD, in presence of suitable mixing sequences, can heavily perturb spins over wide range of resonance frequencies.

## 2  Materials and methods

All experiments have been performed on a Bruker NMR spectrometer in a field of 14.1 T (600 MHz proton frequency) equipped with a probe cooled by liquid nitrogen ("Prodigy") with coils to generate pulsed field gradients along the $z$-axis. This study has been done on a standard calibration sample that contained, among other substances, about 80% $H_2O$ and 20% HDO (*i.e.*, close to 100 M solvent protons) and 0.5 mM Sodium-trimethyl-silyl-propane-sulfonate (DSS). At the experimental temperature of 298 K, the chemical shift difference between the solvent and the methyl protons is *ca.* 4.78 ppm (2868 Hz at 14.1 T, the water resonance being "downfield", i.e., precessing at a higher negative frequency).

The variants of the selective TOCSY experiment (Davis and Bax (1985); Kessler et al. (1986)) used in this work, with an optional bipolar gradient pair for coherence pathway selection (Dalvit and Bovermann (1995)), are described in figure 2. A selective pulse applied to the solvent $A$, followed by a pulsed field gradient, can be inserted before the sequence so that the magnitude of the longitudinal magnetization $M_z^A$ can be controlled and hence the strength of the RD effect. If, for example, the pulse rotates the magnetization into the $xy$-plane, RD should play no role in the remainder of the sequence, except if the dephased magnetization is (accidentally) refocused. If, instead of the transverse magnetization, one wishes to monitor the $z$-component of the magnetization that remains after the homo-nuclear mixing sequence, a gradient followed by a $\pi/2$ pulse can be inserted just before acquisition. For homonuclear transfer, an isotropic mixing pulse train, DIPSI-2 (Rucker and Shaka (1989)), has been chosen with an RF amplitude $\gamma B_1/2\pi = 4.17$ kHz (which corresponds to a duration of 60 $\mu$s for a $\pi/2$ pulse). Selective excitation, either on the water or on the methyl protons, has been achieved with a Gaussian $\pi/2$ pulse of 5 ms.

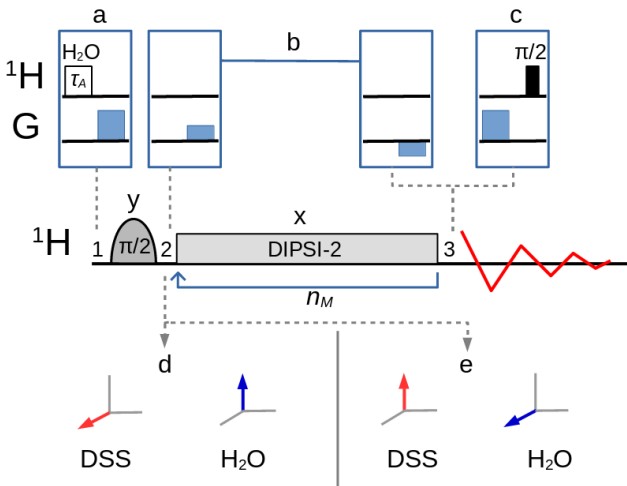

**Figure 2.** Selective TOCSY sequence. The magnetization of one nuclear spin species is rotated into the transverse plane by the selective $\pi/2$ pulse, followed by a DIPSI-2 pulse train which is repeated $n_M$ times. Neglecting relaxation and coherence transfer, the isotropic mixing DIPSI-2 sequence is designed to leave the magnetization unchanged (spin-locked) across a wide band of frequencies centered on the RF carrier frequency. The selective pulse is cycled through $(y, -y, -y, y)$ with a concomitant alternation of the receiver phase. (a) A selective pulse of duration $\tau_A$ applied to the water resonance followed by a pulsed field gradient can be inserted at position 1 to adjust the amplitude of the longitudinal components of the water magnetization between $+M^{eq}$ and $-M^{eq}$. (c) At position 3, a pulsed field gradient followed by a $\pi/2$ pulse permits the detection of $M_z$. (b) An optional bipolar pulsed field gradient pair at positions 2 and 3 on each side of the mixing interval leads to a cleaner coherence pathway selection and a higher signal-to-noise ratio if the receiver gain can be increased, albeit at the cost of some signal decay due to translational diffusion (in addition, the gradient delays of about 3 ms cause a small loss due to transverse relaxation). In this work the carrier frequency for all pulses was set either on the three methyl group resonances of DSS (leading to the situation -immediately after the selective $\pi/2$ pulse- shown in panel (d)) or on the water resonance (e).

The programs for numerical simulations of the trajectories of the magnetization (see supporting information for the code) and to extract the experimental peak intensities were written in the Python language. In particular, the evolution of the magnetization under the DIPSI-2 pulse train (governed by the set of non-linear coupled differential equations 1-3) was numerically evaluated with the SciPy integration libraries (Virtanen et al. (2020)) using an explicit Runge-Kutta method of order 5 (RK5(4)) (Shampine (1986)).

## 3   Experimental results

First, the selective TOCSY experiment of figure 2 was applied with the RF carrier frequency set on the protons of the three methyl groups of DSS. The isotropic mixing module, DIPSI-2, consists of 36 RF pulses of constant amplitude and varying

Number of mixing cycles $n_M$

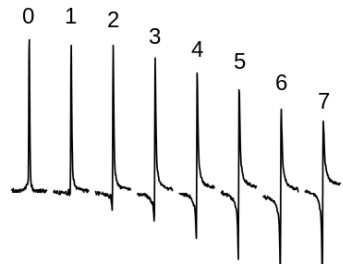

0  1  2  3  4  5  6  7

$\tau_A$ [ms], pulse applied on water (Fig. 2a)

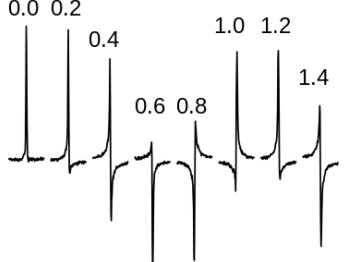

0.0  0.2  0.4  0.6  0.8  1.0  1.2  1.4

**Figure 3.** Spectra of the protons of the three methyl groups of DSS obtained with the experiment of figure 2 with the carrier set at the methyl resonance frequency (conditions before mixing as in figure 2d). The selective $\pi/2$ pulse had a Gaussian profile of 5 ms. The strength of the RF amplitude during mixing was 4.17 kHz. (left) As the number of cycles $n_M$ increases, the signal changes phase. Each pulse train cycle takes about 7 ms to complete. After 26 cycles the resonance is back close to its initial phase (corresponding to a precession frequency close to 5.5 Hz). (right) The amount of $z$-magnetization of $H_2O$ is varied by applying a rectangular pulse with 250 Hz amplitude and a length $\tau_A$ (marked on top of each spectrum) to the water resonance followed by a pulsed field gradient (figure 2a) immediately before the sequence with $n_M = 26$. All spectra have the same phase corrections.

duration, applied along $+x$ or $-x$ and is repeated $n_M$ times (Rucker and Shaka (1989)). Since the excited methyl spins $S$ are not coupled, the mixing sequence acts as a spin-lock and only a decay due to relaxation should be observed as $n_M$ increases. Nevertheless, the spectra of figure 3 (left) show a clear phase-drift, making nearly a full turn at $n_M = 26$. The change of phase

depends strongly on the water $M_z^A$ magnetization at the beginning of the experiment, as can be seen on the right of the figure: for $n_M = 26$, immediately before the selective TOCSY sequence, an RF pulse applied to the $H_2O$ resonance of varying length $\tau_A$, followed by a gradient, has been inserted, so as to modify at will $M_z^A$ before the isotropic mixing sequence.

In figure 4 (left), the phase variations of the latter experiment are plotted as a function of $\tau_A$. Clearly, the magnitude of water magnetization that is present before the pulse sequence modulates the effect observed. The theoretical curve in orange predicts

the phase evolution assuming that the phase is proportional to the initial longitudinal water magnetization, $M_z^A$, $and$ the RF pulse on the water is ideal (i.e., with a nutation angle equal to $\omega_1 \tau_A$). The deviations between the curve and the experimental points could be due to RF inhomogeneities, RD during the pulse applied to water, slight miss-calibrations of the RF power and a possible small miss-estimation of the initial phase-shift. Moreover, RD being a non-linear phenomenon, it is not $a\ priori$ clear that the theoretical curve should be followed. At positions $a$ ($\tau_A = 0$ ms, when the water magnetization is unperturbed),

$b$ ($\tau_A = 1.1$ ms, when the water magnetization approximately vanishes) and $c$ ($\tau_A = 2.2$ ms, when the water magnetization is approximately inverted) the phase evolution has been recorded as a function of number $n_M$ of isotropic mixing cycles, as shown (red dots) on the right of figure 4. The dashed curve corresponds to a linear regression of the first half of the points of $a$, showing that a larger $n_M$ the dephasing slows down slightly (due to relaxation of the water magnetization). When a bipolar

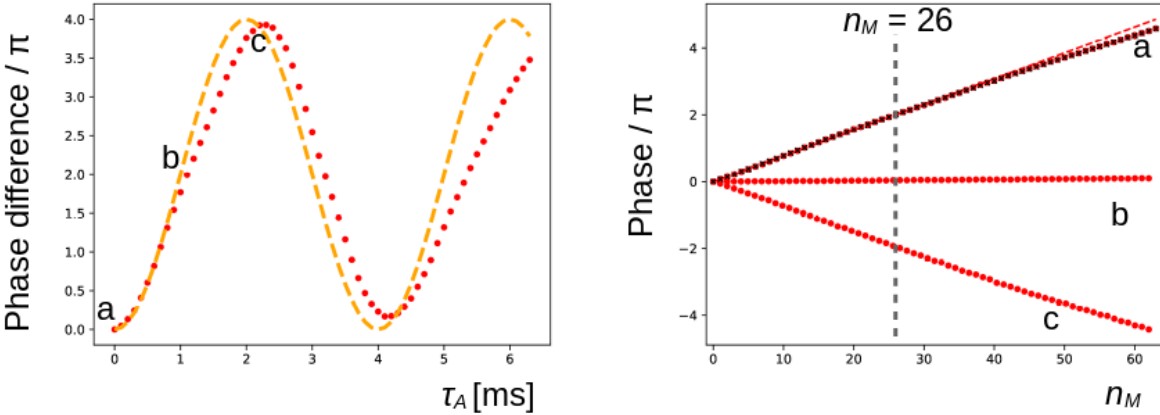

**Figure 4.** (left) The phase evolution of the methyl signal of DSS extracted from the experiment shown on the right side of figure 3 ($n_M = 26$). The duration of the preparatory pulse applied on $H_2O$ has been varied from 0 to 6.3 ms with increments of 0.1 ms (maximum nutation angle of $ca.$ $3\pi$). The orange dashed line shows the expected variation for an ideal RF pulse. (right) At positions $a$ (0 ms), $b$ (1.1 ms, no residual solvent magnetization) and $c$ (2.2 ms, inversion) the phase evolution of the signal is shown (red dots) for $0 \leq n_M \leq 63$ with increments of 1. The red dashed line corresponds to a linear fit of the first 32 points of $a$. The black crosses are recorded under the same conditions of $a$ after inserting a bipolar gradient pair before and after the mixing period as explained in figure 2 and are virtually undistinguishable from the red dots underneath.

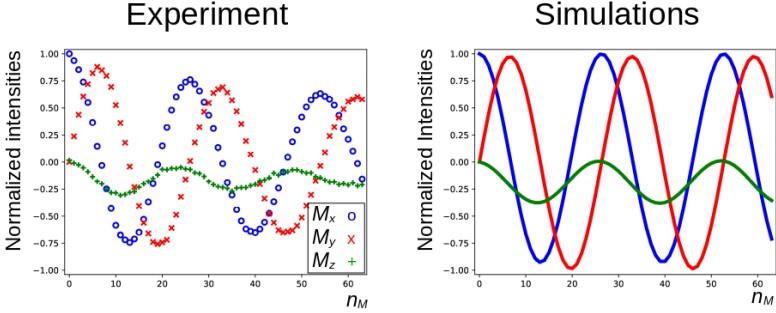

**Figure 5.** Evolution of the magnetization of the methyl resonance of DSS under the experimental conditions of figure 4 (curve $a$). For the calculations on the right the RD rate and angle were set to $R_R = 33.4 \times 2\pi$ rad/s and $\psi = 30°$.

gradient pair is inserted to bracket the DIPSI-2 mixing sequence (black crosses) the effect of RD on the DSS resonance is
almost undistinguishable from the same experiment that does not use gradients for coherence pathway selection.

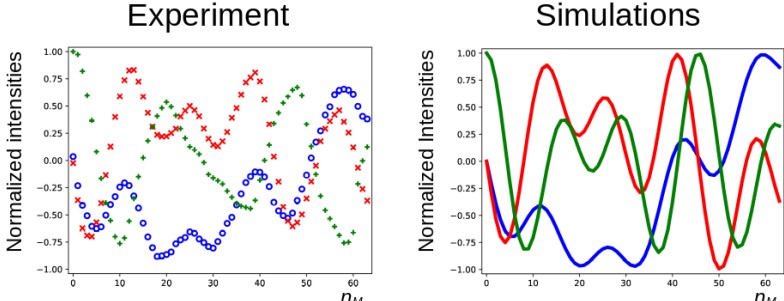

**Figure 6.** Same as figure 5, with the carrier frequency set on $H_2O$ (conditions before mixing as in figure 2e), the same symbols as in figure 5 correspond to the same components of the magnetization. The RD parameters for the simulations on the right were the same as in figure 5.

In figure 5 (left), the three components of the magnetization of the DSS methyl groups, recorded under the same conditions as figure 4 (curve $a$), are plotted as a function of $n_M$. Figure 6 (left) shows the result of an experiment, where the carrier frequency has been moved to the solvent resonance and the amplitude of the selective Gaussian pulse has been increased in order to overcome RD effects during this pulse (so that the solvent magnetization is rotated into the $xy$-plane). All other parameters
were left unchanged. Here, the residual $z$-component of the magnetization of DSS must be detected without changing the phase of the receiver for the different scans (the initial magnetization and the detected component have the same coherence order). Without RD, the magnetization of the methyl groups is expected to stay along the $z$-axis. Clearly, effects of the RD field are also observed in the latter experiment. Without RD the magnetization is expected to stay along the $z$-axis after each DIPSI-2 cycle. As in figure 5, the magnetization rotates away from its initial position, although its trajectory is much less regular.

**4 Theory and discussion**

In order to explain the experimental results, the homo-nuclear case of abundant spins $A$ ($H_2O$), whose magnetization induces an RD field in the coil as shown in figure 1, and sparse spins $S$ (the three methyl groups in DSS), whose RD interaction with the coil can be neglected, will be considered. In the rotating frame, the evolution of the two (uncoupled) types of spins can be described by the modified Bloch equations (Bloom (1957)):

$$\frac{dM_x^i(t)}{dt} = -\omega_0^i M_y^i(t) + \omega_{1y}(t)M_z^i(t) - \{c_{Rx}(t) - s_{Ry}(t)\}M_z^i(t), \tag{1}$$

$$\frac{dM_y^i(t)}{dt} = \omega_0^i M_x^i(t) - \omega_{1x}(t)M_z^i(t) - \{c_{Ry}(t) + s_{Rx}(t)\}M_z^i(t), \tag{2}$$

$$\frac{dM_z^i(t)}{dt} = -\omega_{1y}(t)M_x^i(t) + \omega_{1x}(t)M_y^i(t)$$
$$+ \{c_{Rx}(t) - s_{Ry}(t)\}M_x^i(t) + \{c_{Ry}(t) + s_{Rx}(t)\}M_y^i(t), \tag{3}$$

with $i$ either spin $A$ or $S$, $\omega_0^i$ the difference between the resonance frequency of spin $i$ and the carrier frequency, $\omega_{1x}$ and $\omega_{1y}$ the $x$ and $y$ components of the RF field during the mixing sequence, while the remaining terms in the equations are due to the
RD field:

$$c_{Rx}(t) = \alpha_R M_x^A(t)\cos(\psi), \ s_{Rx}(t) = \alpha_R M_x^A(t)\sin(\psi),$$
$$c_{Ry}(t) = \alpha_R M_y^A(t)\cos(\psi), \ s_{Ry}(t) = \alpha_R M_y^A(t)\sin(\psi), \tag{4}$$

where the amplitude of the RD field is $\omega_R(t) = \alpha_R\sqrt{M_x^A(t)^2 + M_y^A(t)^2}$ and its phase is given by the angle $\psi$ as indicated in figure 1. The proportionality constant $\alpha_R$ depends on the characteristics of the RF circuit:

$$\alpha_R = \cos(\psi)\mu_0\eta\gamma Q/2 \,, \tag{5}$$

with $\mu_0$ the vacuum permeability, $\gamma$ the gyro-magnetic ratio of the protons, $\eta$ the filling factor of the sample and $Q$ the quality factor of the RF circuit. By a multiplication of $\alpha_R$ with the equilibrium magnetization of the abundant spins $A$, the use of the RD rate

$$R_R = \alpha_R M_{eq}^A \,, \tag{6}$$

allows one to use normalized magnetization vectors (*i.e.*, divide all components of spin $i$ by $M_{eq}^i$) in equations 1-4.

The evolution of the magnetization of both $A$ and $S$ nuclei during the DIPSI-2 pulse train has been numerically simulated using the above equations. First the evolution of $\boldsymbol{M}^A(t)$ was determined. For spin $S$ equations 1-3 reduce to the traditional Bloch equations, with the magnetization of spin $A$ as a source of a time-dependent RF field. The values of the rate $R_R$ and the angle $\psi$ were estimated (the two values have independently been varied in the simulations) to give a qualitative agreement with the data, as shown in figure 5, rather than an exact fit. The combination of the two parameters is not unique, a smaller angle
$\psi$ can be compensated by a larger value of $R_R$. The use of the RD parameters extracted from the signal of $H_2O$ after a simple pulse-acquire experiment does not lead to a good agreement. This is likely due to the fact that the RF circuit is not the same during signal acquisition as during the application of RF pulses (Marion and Desvaux (2008); Pöschko et al. (2014)). In figure 5, the agreement between simulations and experiments is quite satisfactory. The decay of the experimental curves is not only due to relaxation but also to RF inhomogeneities: the precession frequency of the DSS signal varies slightly with the the RF
amplitude, while the evolution of the $z$ component is even more sensitive (see supporting information). The perturbation is also present when the carrier during the mixing is set on a frequency different from DSS resonance. In the supporting information, simulations are presented for different offset frequencies.

     For the curves on the right-hand side of figure 6 the same RD parameters in figure 5 have been used. In the simulations, the fact that, due to RD effects, the water magnetization is not aligned along the $x$-axis after the first $\pi/2$ pulse has been taken

into account (a phase shift of -18° was determined experimentally). The agreement between experiments and simulations is adequate, considering the fact that neither RF inhomogeneity and calibration errors nor relaxation effects have been taken into account. Moreover, the evolution is very sensitive to the exact position of the water magnetization after the selective Gaussian pulse.

The deviation from quadrature between the RD field and the transverse magnetization of the solvent ($\psi \neq 0$) plays an
important role. It can be particularly pronounced in cryogenically cooled probes (Shishmarev and Otting (2011)). Previous studies have shown the influence of this "non-ideality" on the evolution of the magnetization of the abundant spins. Sleator and coworkers (Sleator et al. (1987)), Vlassenbroek and coworkers (Vlassenbroek et al. (1995)), and Torchia (Torchia (2009)) demonstrated a polarization dependent phase-shift of the freely precessing signal, while Barjat and coworkers (Barjat et al. (1995)) brought to light severe phase distortions in multiplets. Williamson and coworkers (Williamson et al. (2006)) revealed
that it causes asymmetries in $Z$-spectra, potentially perturbing the observation of chemical exchange. It is instructive to investigate what happens with a far stronger RF amplitude, which much reduces possible imperfections of the DIPSI-2 sequence due to offset effects. In figure 7, several simulations of the trajectory of the DSS magnetization are presented with an amplitude of the RF field of the spin-lock of 100.0 kHz. The carrier frequency was set on the water resonance and the maximum mixing time was 0.200 s. Different initial conditions (shown above each graph) before the DIPSI-2 pulse train have been considered.
When the initial magnetization of $H_2O$ is aligned along one of the principal axes, the one of DSS nutates around this axis, while when this is not the case (lower right corner), the trajectory of the DSS magnetization is more complex. When the initial solvent magnetization is put along the $x$-axis, it stays *continuously* almost perfectly aligned along this axis and (neglecting relaxation) the RD field is constant, with one component parallel to the magnetization equal to $\omega_{Rx} = R_R \sin(\psi)$ and one term perpendicular to it along the y-axis equal to $\omega_{Ry} = -R_R \cos(\psi)$. The latter component is efficiently (in analogy to the offset
term) averaged out from the effective Hamiltonian by the DIPSI-2 sequence, while the first component is unaffected since it commutes with the RF field at all times. Hence, the effect of the RD field on the dilute spins (after each completed DIPSI-2 cycle) is expected to be a nutation around the $x$-axis with a frequency of $R_R \sin(\psi)/2\pi$. Indeed, this corresponds to the frequency of 16.7 Hz found in the graph on the top left corner. When the initial solvent magnetization is perpendicular (either along the $z$- or the $y$-axis) to the spin-lock field, the RD field becomes time-dependent. The net effect is a nutation around the
axis of the initial solvent magnetization with a frequency of about one half compared to the previous one (in the appendix it is demonstrated that this is to be expected). When the initial magnetization of the solvent is oriented arbitrarily, the trajectory is less regular. The evolution of the solvent magnetization can also be strongly influenced by the much weaker RD field when the orientation of the initial magnetization is not along one of the three main axes (in the supporting information, the trajectories of the solvent magnetization under the same conditions as in figure 7 are shown).

In the previous simulations, relaxation has not been taken into account. Since it causes the magnetization of the solvent to diminish, its effect on the remote resonances should become weaker as the number of spin-lock cycles increases (as the regression line in figure 4 shows). In the supporting information, simulations with several different values of the transverse relaxation rates of the solvent $R_2^A$ illustrate this effect.

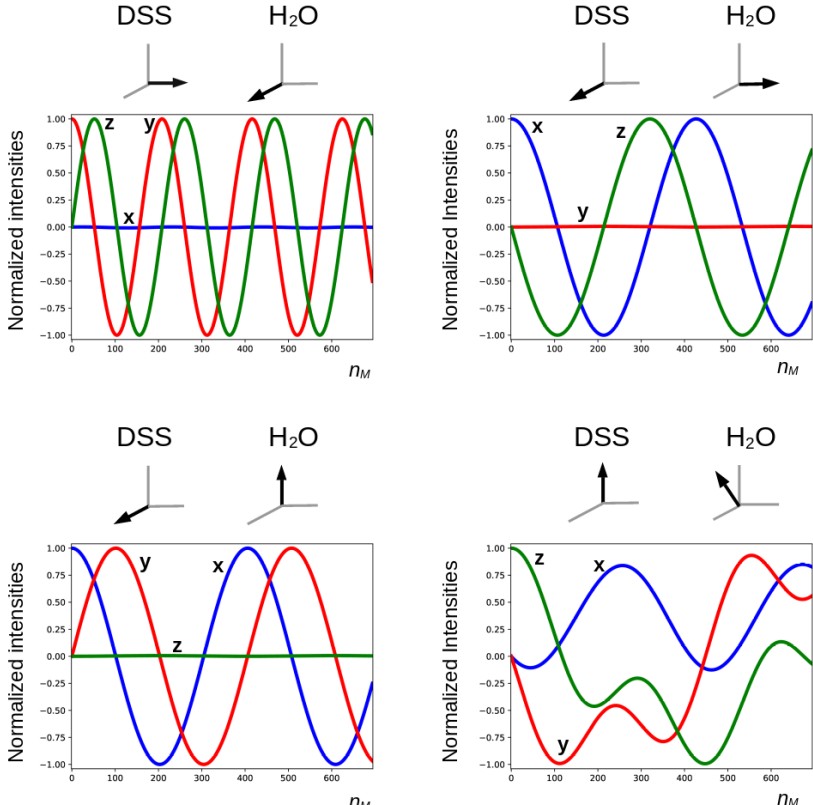

**Figure 7.** Simulated trajectories of the magnetization of the methyl groups of DSS under a DIPSI-2 pulse train with an RF amplitude of 100 kHz (90° pulse of 2.5 $\mu$s), with the carrier frequency set on the water resonance. On top of each graph, the initial magnetization just before the spin-lock is shown (on the lower right corner $M_x^A(0)/M_{eq}^A = 0.411$, $M_y^A(0)/M_{eq}^A = -0.310$, $M_z^A(0)/M_{eq}^A = 0.857$). The RD parameters used for the simulations and the color code of the curves are identical to the ones used in figures 5 and 6. For clarity, the curves are also labeled with the corresponding direction of the magnetization. The maximum number of cycles $n_M = 694$ corresponds to a duration of 0.200 s.

In this work, the selective TOCSY experiment has been investigated. For a (non-selective) two-dimensional TOCSY experiment the situation is more complex since, due to chemical shift evolution and RD, the orientations of the different magnetization vectors just before mixing depend on the duration of the indirect evolution period. In the supporting information, simulations are shown where the magnetization of both resonances is aligned along the $x$-axis before the mixing period (effect much

smaller) or where the magnetization of the water is aligned along the $y$-axis while the one of DSS is along the $x$-axis (effect comparable to figure 5).

The effect described in this article, which has been demonstrated using the DIPSI-2 mixing sequence, is expected to be present in other sequences which efficiently remove the chemical shift from the effective Hamiltonian. In the supporting information, simulations show that this is indeed the case for the MLEV-16 (Levitt et al. (1982)) and the FLOPSY-16 (Kadkhodaie et al. (1991)) sequences. In contrast, a continuous wave spin-lock with the same RF amplitude does not exhibit this effect.

The phenomenon shown in this work strongly depends on the characteristics of the probe. Similar results (not shown), albeit much smaller in magnitude, have been obtained at 18.8 T (800 MHz proton frequency) on a traditional "room temperature" probe.

## 5 Conclusions

It has been shown that, in presence of sequences used for TOCSY mixing, RD can strongly perturb the evolution of magnetization of spins that are neither directly coupled by scalar or dipolar interactions to the source spins nor have a nearby resonance frequency. Since this type of mixing sequences efficiently removes the chemical shift differences from the effective Hamiltonian, the weak RD field affects resonances over a much wider range of frequencies than would be expected from its amplitude. Counter-intuitively, the RD field can thus cause the magnetization of remote resonances to precess notwithstanding the presence of a much stronger RF spin-locking pulse-train. This effect increases with increasing RF amplitudes. Its magnitude depends on the instrumentation, the details of the pulse sequence and the duration of the mixing time. It can be prevented by saturating or dephasing the magnetization of the spins that cause radiation damping before mixing.

## 6 Appendix: evolution of the passive spins under a strong DIPSI-2 field and a time-dependent RD field

The basic element of the DIPSI-2 sequence consists of 9 pulses along the $x$-axis of constant amplitude and alternating in sign with the following rotation angles $\phi_n$ for spins on resonance:

$$R = 320_x 410_{-x} 290_x 285_{-x} 30_x 245_{-x} 375_x 265_{-x} 370_x \ . \tag{A1}$$

One full mixing cycle consists of a sequence:

$$R\bar{R}\bar{R}R \ , \tag{A2}$$

in which $\bar{R}$ denotes the same sequences of pulses, except for an opposite sign of the amplitude. The conditions of figure 7 are considered, i.e. the RF term of the Hamiltonian much larger than the other terms and the carrier frequency set on the resonance of the solvent. First, the situation on the top-right corner of figure 7 will be looked at. The trajectory of the solvent magnetization is considered to be governed only by the dominant RF field, which means that it turns around the $x$-axis and the RD field is entirely due to the $y$-component of the solvent magnetization. For a spin $I$, during the $k^{th}$ pulse, the time-dependent

Hamiltonian is thus given by:

$$H_{Ik}(t) = \{s_k|\omega_1| + R_R\cos(\psi)\cos(\Phi_k + s_k|\omega_1|t)\}I_x + R_R\sin(\psi)\cos(\Phi_k + s_k|\omega_1|t)I_y + \omega_0^I I_z , \tag{A3}$$

with:

$$\Phi_k = \sum_{j=0}^{k-1} s_j\phi_j \tag{A4}$$

where $s_0 = 1$, $s_n$ the sign of the RF amplitude of the $n^{th}$ pulse, $\phi_0 = 0$, $\phi_n$ the absolute value of the rotation angle of the $n^{th}$ pulse and $t$ is the time starting from the beginning of the pulse. The offset term causes a negligible tilt of the RF field and is efficiently averaged out by one complete DIPSI-2 cycle and can be omitted in the further analysis. The time-dependent term that depends on $I_x$ commutes with the RF Hamiltonian and averages out. It will also be left out (this is not completely rigorous and can only be done if $|R_R t| << 1$, hence the higher the RF amplitude the better the approximation). Consequently, equation A3 becomes:

$$H_{Ik}(t) = a_k I_x + b\cos(\Phi_k + a_k t)I_y , \tag{A5}$$

with $a_k = s_k|\omega_1|$ and $b = R_R\sin(\psi)$. By going to a rotating (around the $x$-axis) frame with an angular frequency of $a_k$, neglecting the counter rotating component and tilting the frame by an angle $\Phi_k$ (also around the $x$-axis), the Hamiltonian can be expressed as:

$$H_{Ik}^{rt}(t) = bI_y/2 . \tag{A6}$$

The direction of the rotation depends on the sign of the RF amplitude. In its original frame the propagator of the $k^{th}$ pulse is thus given by:

$$P_{Ik} = e^{-is_k\phi_k I_x}e^{-i\Phi_k I_x}e^{-i\tau_k bI_y/2}e^{i\Phi_k I_x} , \tag{A7}$$

where $\tau_k = \phi_k/|\omega_1|$ is the duration of the $k^{th}$ pulse. Since $\Phi_k + s_k\phi_k = \Phi_{k+1}$, the propagator of one full DIPSI-2 cycle (obtained by concatenating the propagators of all pulses) is:

$$P_{ID}^y = e^{-is_K\phi_K I_x}e^{-i\Phi_K I_x}e^{-i\tau_D bI_y/2}e^{i\Phi_1 I_x} = e^{-i\tau_D bI_y/2} , \tag{A8}$$

where $K$ (=36) stands for the last index of the cycle, $\tau_D$ is the total duration of the cycle, and the latter equality is because $\Phi_1 = \phi_0 = 0$ and $\Phi_K + s_K\phi_K = 0$ (ensured by equation A2). Since this propagator leaves the initial solvent magnetization unchanged, it stays identical for subsequent cycles and the effective Hamiltonian is equal to $R_R\sin(\psi)I_y/2$.

When the initial solvent magnetization is aligned along the $z$-axis, the only difference is that the angle $\phi_0 = 90°$, hence equation A8 becomes:

$$P_{ID}^z = e^{-i\pi I_x/2}e^{-i\tau_D bI_y/2}e^{i\pi I_x/2} = e^{-i\tau_D bI_z/2} . \tag{A9}$$

The nutation frequencies observed in figure 7 slightly deviate from the expected ones (about -2% for the graph in the top right, and +2% for the lower left corner). This is likely due to effects of the neglected counter rotating component, since this effect persists even when removing the offset and the RD term parallel to the RF field from the Hamiltonian in equation A3. This component is rapidly modulated by the changes of sign of the RF amplitude between the pulses, so that it could affect a wider range of frequencies than can be expected from its amplitude. For other orientations (perpendicular to the RF field, but not along either of the two principal axes) of the initial solvent magnetization, these perturbations do slightly modify the orientation of the solvent magnetization after each cycle, causing non-linear effects and ever larger deviations with increasing number of cycles.

When the carrier frequency is not set on the solvent resonance frequency, the analysis is more complex. In the supplementary information, simulations similar to the ones in figure 5 are shown (carrier frequency set on the DSS resonance frequency) with varying RF amplitudes.

*Author contributions.* $N.A$

*Competing interests.* The author has no conflicts of interest to declare.

*Acknowledgements.* I thank Vineeth Francis Thalakottoor for help with the simulations, Geoffrey Bodenhausen for correcting the manuscript and Tom Barbara and Matt Augustine for insightful comments.

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
