# Peer review of "Radiation damping strongly perturbs remote resonances in presence of homo-nuclear mixing"

_Magnetic Resonance, 2021_

## Referee Comment (RC1)

*REVIEWER REPORT on "Radiation damping strongly perturbs remote resonances in presence of homo-nuclear mixing sequences"*

*GENERAL IMPRESSION: A very interesting paper, which should be revised somewhat to gain even more impact.*

- *Does the paper address relevant scientific questions within the scope of MR?*
Yes, the paper addresses an interesting experimental effect of radiation damping on remote resonances in spin lock experiments. It may also offer a clue for the physical reason phasing problems often encountered in biomolecular NMR experiments in aqueous solution involving TOCSY-type coherence transfer, although the author does not mention this and is maybe not aware of it.

- *Does the paper present novel concepts, ideas, tools, or data? All submitted papers are assumed to report on new observations and/or new theory; there is no need to draw attention to the novelty in title, abstract, or conclusions.*

Yes the concept is new and has to my knowledge not been reported earlier.

- *Are substantial conclusions reached?*
Yes, but (as mentioned above) there may be even more to it than the author reveals.

- *Are the results sufficient to support the interpretations and conclusions? Is the description of experiments and calculations sufficiently complete and precise to allow their reproduction by fellow scientists with reasonable effort? Detailed technical and graphical explanations and documentation of limited file size can be provided as supporting information. Access to raw data, processed spectra, and other experimental data must be provided by depositing in a publicly accessible repository or archive as far as practically feasible, and the DOI provided in the article. Hardware developments need to be documented by photos or equivalent drawings (blueprints with precise dimensions if possible). New software must be accompanied by user instructions. New software should be open source and access to it provided through a software repository if possible.*

The experiments are clearly described. The formulae and symbols are clearly explained. However, it is not clear to me, how exactly the simulation was done. It appears that an "ideal spin lock" was simulated not the actual DIPSI pulse sequence used. But that is more from reading "between the lines". It would be beneficial to provide the simulation code in a Supplementary Information document.
Some questions remain open, which might at least be addressed in the discussion: Influence of relaxation times, e.g. a solvent with a long T2 like acetone, the importance of the type of the mixing pulse sequence (planar mixing vs isotropic).
What the effect be for a non-selective excitation pulse?
Does the imaginary component of RD, which causes tuning and polarization dependent phase shifts (Torchia(2009): DOI:10.1007/s10858-009-9363-6), have an influence on the effects observed under spin-lock conditions? What is the offset dependence, in theory and in practice.

• *Are numerical data accompanied by error estimates with a description of the methods used to obtain these estimates?*

There are no error estimates given, but in the pertinent context addition of error estimates would not add much to the impact and scientific value of the paper.

• *Do the authors give proper credit to related work and clearly indicate their own new/original contribution?*

The papers quoted are adequate. In my detailed comments below, I suggest a few additional references.

• *Does the title clearly reflect the contents of the paper?*

In principle yes. But, as only one mixing sequence is used in the paper I suggest to drop the last word of the title.

• *Does the abstract provide a concise and complete summary?*

The abstract is very short. It uses the term "inductive coupling", which is not occurring anywhere else in the text (where the term "inductive backaction" is introduced). I'd suggest to use the term from the abstract also in the text. Again there is the plural "sequences" but only one sequence is used in the experiments.

• *Is the overall presentation well-structured and clear?*

Yes, except for the two problems mentioned above (what exactly was simulated; do the results also apply to other mixing sequences).

• *Is the language fluent and precise?*

Yes.

• *Should any parts of the paper (text, formulae, figures, tables) be clarified, reduced, combined, or eliminated?*

See my detailed comments to the figures. Figures 5 and 6 should be combined into one figure with four panels.

• *Are the number and quality of references appropriate?*

See detailed comments in the detailed comments.

• *Is the amount and quality of the supporting information and supplementary material appropriate?*

A supplementary file should be added, see comments in the detailed comments below.

DETAILED COMMENTS:

P.1
Title:
Actually only a single sequence is being used in this paper, so "sequences" should be dropped from the title

Abstract:
use "inductive coupling" also in the main text
avoid "sequences", se above

Figures in general:
In most figures the light colors are too light, some lines too thin, please improve the presentation.

Figure 1:
the text label colors should match the graphical elements they refer to.

Figure Caption 1:
use the symbols from the figure in the caption:

RD field $\omega_R$

water magnetization $M^{H2O}$ ..

p.2 li17:
„if it is homogeneous in space, as with any RF field"
It is not clear to what this refers, RF-fields can be inhomogeneous

p.2 li18:
The importance of the quality factor Q should also be mentioned here

p.2 li19:
"partially- or non-deuterated solvents"
maybe better to use something like "highly protonated solvents (at thermal polarization levels)"; as there are solvents without any hydrogen which would also classify as non-deuterated

p.2 li20-25:
A more recent paper describes the interference of solvent RD with small partially overlapping peaks: Schlagnitweit et al. doi: 10.1002/cphc.201100724

p.2 li34: better use "variants of the TOCSY experiment" or similar

p.2 li39: a smaller pulse angle might help to reduce RD effects during the direct detection

p.2 li40: It would be interesting to know the influence of different types of mixing sequences (planar vs. isotropic)

p.3: Fig.2 needs to be improved, in particular the this lines in d and e

Figure 2, caption li.5:
The use of the word "tune" in this context is unusual, use "adjust" or "control"

Figure 2, caption li.10: T2 relaxation losses will also increase due to the additional delays; "Carrier frequency" applies to which pulses?

p.3 li 46: It is not clear to what extent the particular mixing sequence was simulated. The simulation code should be published in a Supplementary Information file or deposited.

p.2 li 48: I'd insert a "First," at the beginning of the section.

p.3 li 51: "relaxation-induced decay" → "decay owed to relaxation" (induce implies some „active" role)

p.4 Fig.3: Suggestion: name the two graphs  two panels a and b instead of left and right

Showing the full range of $n_M$ values  could be instructive (maybe in the Supplementary Info)

p.4 li57: if → assuming
        "water longitudinal" → "longitudinal water"

p.4 li66: The data should be shown in the Supplementary Info

p.4 li68: "…shows the result of an identical experiment, except that the carrier frequency has been moved to the solvent resonance…"

→

"…shows the result of an experiment, where the carrier frequency has been moved to the solvent resonance, and the amplitude of the selective Gaussian pulse has been increased in order to overcome RD effects during this pulse, so that the solvent magnetization is rotated in the xy-plane, while all other parameters wer unchanged." But maybe it would be better to split that long sentence.

Fig.4-6: Combining the three figures into one with 6 panels a,b,c,d,e,f is recommended.

p.5 Fig.4: the "black crosses" are hardly resolved

p.5 Fig.4 caption li.2: "varied" → "was varied"

p.5 Fig.4 caption li. 3: "complete saturation": As the state is reached by a 90° pulse, one should not call it saturation.

p.5. Fig. 5 caption:  How was the RD rate "estimated"?

p.5 li. 70: …rotated to… → …rotated into…

It's not clear what the following sentence means: "Here, the z-component of the magnetization must be detected without changing the phase of the receiver for the different scans." Probably the phase cycle is different as the coherence pathway has been changed. More details should be discussed in that paragraph. "Clearly, effects of the RD field are also observed in the latter experiment." is not sufficient.

p.6: The theoretical approach is presented clearly, except for the fact that there is no explanation of how the particular spin lock pulse sequence was taken into account. One might also introduce definitions of eqs. 4 and 5 before eq.1.

p.7 li96ff: The estimation of the RD rate might be better via a small flip angle or a spin noise experiment. For short T2 a separate determination of T2 under non-RD conditions may be required for correction.

More recent papers elaborating on the differences in probe tuning under receive- and pulse-conditions by Pöschko et al., which might be relevant here: DOI:10.1002/cphc.201402236 (2014) and partially also relevant: DOI:10.1038/ncomms13914 (2017)

*"The decay of the experimental curves is not only due to relaxation but also to RF inhomogeneities: the precession frequency of the DSS signal varies slightly with the  RF amplitude, while the evolution of the z component is even more sensitive (simulations not shown)."*

Please show those simulations in the supplementary information.

---

## Author Comment (AC4)

Dear reviewer,

thanks for your careful reading of the manuscript and your comments. I have addressed the different points in a revised version of the manuscript as described below. I followed also your suggestion to add supporting information.

● *However, it is not clear to me, how exactly the simulation was done. It appears that an "ideal spin lock" was simulated not the actual DIPSI pulse sequence used. But that is more from reading "between the lines". It would be beneficial to provide the simulation code in a Supplementary Information document.*
**In fact, the actual DIPSI-2 sequence has been simulated and not "an ideal spin lock". As the reviewer suggests, the code is now in the supplementary information.**

● *Some questions remain open, which might at least be addressed in the discussion: Influence of relaxation times, e.g. a solvent with a long T2 like acetone*
**Relaxation is now discussed at the end of the discussion section. Simulations with different $T_2$ values have been added to the supporting information.**

● *What the effect be for a non-selective excitation pulse?*
***2D TOCSY experiments are now briefly discussed and simulations for a non-selective excitation pulse have been added to the supporting information.***

● *Does the imaginary component of RD, which causes tuning and polarization dependent phase shifts (Torchia(2009): DOI:10.1007/s10858-009-9363-6), have an influence on the effects observed under spin-lock conditions?*
**The work of Torchia develops upon the article of Vlassenbroek, Jeener and Broekaert (1995). Indeed, the effect described in my article depends much on the fact that the RD field is not in perfect quadrature with the transverse magnetization. The $M_z$ dependent frequency shift described by Torchia is due to the fact that the transverse component stemming from the rotation of $M_z$ by the RD field is not perfectly aligned with the transverse magnetization that causes RD and not because there is a polarization dependent precession frequency (as is the case for the dipolar demagnetizing field). I have added a discussion (see also my answers to the other reviewers) about this aspect, although I do not see how one can directly relate this effect with the one described in my manuscript, apart that they have a common cause.**

● *What is the offset dependence, in theory and in practice.*
***The effect is somewhat offset dependent. This is now discussed in the main body and a figure is added in the supporting information.***

● *In principle yes. But, as only one mixing sequence is used in the paper I suggest to drop the last word of the title.*
**I dropped "sequences" from the title and the abstract, although now other mixing sequences are simulated (see below).**

● *Abstract: use "inductive coupling" also in the main text avoid "sequences", see above*
**"Inductive coupling" is now employed in the main text.**

● *Figures in general: In most figures the light colors are too light, some lines too thin, please improve the presentation.*
**The lines are thicker in all figures and the colors clearer.**

● *Figure 1: the text label colors should match the graphical elements they refer to.*
*Figure Caption 1: use the symbols from the figure in the caption: RD field ωR water magnetization MH2O*
**I changed figure caption. I did not change the colors of the labels, but indicated which arrow corresponds to what in the figure.**

● *p.2 li17: "if it is homogeneous in space, as with any RF field" It is not clear to what this refers, RF-fields can be inhomogeneous*
**I meant to say that this property was not specific for an RD field (all RF fields conserve the norm if they are homogeneous and relaxation can be neglected). Since this comment is trivial and to avoid confusion, I removed "as with any RF field").**

*p.2 li18: The importance of the quality factor Q should also be mentioned here*
**A sentence has been added: "It increases with higher quality factors $Q$."**

● *p.2 li19: "partially- or non-deuterated solvents" maybe better to use something like "highly protonated solvents (at thermal polarization levels)"; as there are solvents without any hydrogen which would also classify as non-deuterated*
**The sentence has been rewritten:**
**"In liquid state NMR, this effect is usually weak and only noticeable when the magnetization is strong, either for nuclei in molar concentrations with high gyromagnetic ratios (in particular solvents containing hydrogen) or when the polarization is enhanced."**

● *p.2 li20-25: A more recent paper describes the interference of solvent RD with small partially overlapping peaks: Schlagnitweit et al. doi: 10.1002/cphc.201100724*
**Thank you for pointing out this article, which is indeed an excellent example for resonances nearby. I added it to the references.**

● *p.2 li34: better use "variants of the TOCSY experiment" or similar*
**The suggestion of the reviewer has been followed.**

● *p.2 li39: a smaller pulse angle might help to reduce RD effects during the direct detection*
**Since it is the remote resonance which is of interest in this article, there was no need to reduce RD effects during detection. The only problems encountered (and only in some of the experiments) were baseline distortions, which could easily be corrected for.**

● *p.2 li40: It would be interesting to know the influence of different types of mixing sequences (planar vs. isotropic)*
**I have added simulations of other mixing sequences (FLOPSY-16 and MLEV-16) in the supplementary information, showing similar effects. The FLOPSY-16 sequence is non isotropic. Planar sequences are usually heteronuclear or bandselective, and cannot be simulated with the program used in this work. Note that the term planar or isotropic refers to the effective coupling Hamiltonian which is not present in the system under study. For the heteronuclear case, planar mixing sequences of course do not exhibit the effect described in this work. Perhaps, this does neither happen for the homonuclear case. An interesting question (but outside the scope of this work) is the case when a scalar (or residual dipolar) coupling is present between the source and the remote resonance.**

● *p.3: Fig.2 needs to be improved, in particular the this lines in d and e Figure 2, caption li.5: The use of the word "tune" in this context is unusual, use "adjust" or "control"*
**I improved the figure and changed "tune" into "adjust".**

● *Figure 2, caption li.10: T2 relaxation losses will also increase due to the additional delays;
"Carrier frequency" applies to which pulses?*
**The sentence "(in addition, the gradient delays of about 3 ms cause a small loss due to
transverse relaxation)" has been added and I specified that the carrier frequency applies to
all pulses.**

● *p.3 li 46: It is not clear to what extent the particular mixing sequence was simulated. The
simulation code should be published in a Supplementary Information file or deposited.*
**See above. I changed TOCSY for DIPSI-2 in this line.**

● *p.2 li 48: I'd insert a "First," at the beginning of the section.*
**"First," has been inserted.**

● *p.3 li 51: "relaxation-induced decay" à "decay owed to relaxation" (induce implies some
„active" role)*
**It has been changed to "a decay due to relaxation".**

● *p.4 Fig.3: Suggestion: name the two graphs two panels a and b instead of left and right
Showing the full range of nM values could be instructive (maybe in the Supplementary Info).*
**I prefer to keep the figure as is. The full range is in my opinion not more instructive than the
points in the later figures.**

● *p.4 li57: if à assuming "water longitudinal" à "longitudinal water"*
**The text has been changed accordingly.**

● *p.4 li66: The data should be shown in the Supplementary Info*
**The black crosses and red filled circles, corresponding to experiment with and without
gradient selection, are both already shown in the figure. The point is that they overlap. I have
made this point now clearer in the main text and the figure caption.**

● *p.4 li68: "…shows the result of an identical experiment, except that the carrier frequency has
been moved to the solvent resonance…" à
"…shows the result of an experiment, where the carrier frequency has been moved to the
solvent resonance, and the amplitude of the selective Gaussian pulse has been increased in
order to overcome RD effects during this pulse, so that the solvent magnetization is rotated
in the xy-plane, while all other parameters wer unchanged." But maybe it would be better to
split that long sentence.*
**The text has been changed as the reviewer suggests (in two sentences).**

● *Fig.4-6: Combining the three figures into one with 6 panels a,b,c,d,e,f is recommended.*
**I prefer to keep 3 figures.**

● *p.5 Fig.4: the "black crosses" are hardly resolved*
**The figure has been enlarged. It is difficult to resolve all 64 points in particular since there are
two sets of points. However, it is clear that the red points and black crosses overlap (this point
has now been emphasized).**

● *p.5 Fig.4 caption li.2: "varied" à "was varied"*
**"varied" has been changed to "has been varied"**

● *p.5 Fig.4 caption li. 3: "complete saturation": As the state is reached by a 90° pulse, one should not call it saturation.*
**It has been changed to "no residual magnetization".**

● *p.5. Fig. 5 caption: How was the RD rate "estimated"?*
**In the caption "set to" has been used instead of "estimated", in the main text "(the two values have been independently varied)" has been added.**

● *p.5 li. 70: …rotated to… à …rotated into…*
**The correction has been included.**

●*It's not clear what the following sentence means: "Here, the z-component of the magnetization must be detected without changing the phase of the receiver for the different scans." Probably the phase cycle is different as the coherence pathway has been changed. More details should be discussed in that paragraph.*
**The sentence has been clarified in the manuscript.**

●*"Clearly, effects of the RD field are also observed in the latter experiment." is not sufficient.*
**I have extended the description.**

● *p.6: The theoretical approach is presented clearly, except for the fact that there is no explanation of how the particular spin lock pulse sequence was taken into account. One might also introduce definitions of eqs. 4 and 5 before eq.1.*
**First comment: see above. I prefer to keep the order of the equations.**

● *p.7 li96ff: The estimation of the RD rate might be better via a small flip angle or a spin noise experiment. For short T2 a separate determination of T2 under non-RD conditions may be required for correction. More recent papers elaborating on the differences in probe tuning under receive- and pulse-conditions by Pöschko et al., which might be relevant here: DOI:10.1002/cphc.201402236 (2014) and partially also relevant: DOI:10.1038/ncomms13914 (2017)*
**There might be better and more precise methods to estimate the RD rate (although 90° pulse-acquire is not that bad), still they do not give the parameters under RF irradiation. The effect of $T_2$ has now been simulated and put in the supplementary information. I added the first reference which gives a good overview of the differences in probe tuning.**

● *"The decay of the experimental curves is not only due to relaxation but also to RF inhomogeneities: the precession frequency of the DSS signal varies slightly with the the RF amplitude, while the evolution of the z component is even more sensitive (simulations not shown)." Please show those simulations in the supplementary information.*
**The simulations have been included in the supporting information.**

Kind regards,
Philippe